# Inflammatory Cell-Targeted Delivery Systems for Myocardial Infarction Treatment

**DOI:** 10.3390/bioengineering12020205

**Published:** 2025-02-19

**Authors:** Wenyuan Zhang, Dan Peng, Shiqi Cheng, Rui Ni, Meiyang Yang, Yongqing Cai, Jianhong Chen, Fang Liu, Yao Liu

**Affiliations:** Department of Pharmacy, Daping Hospital, Army Medical University, Chongqing 400042, China; 13397085106@163.com (W.Z.); danj1818@tmmu.edu.cn (D.P.);

**Keywords:** inflammation, immune cells, fibroblasts, delivery system, myocardial infarction

## Abstract

Myocardial infarction (MI) is a cardiovascular disease (CVD) with high morbidity and mortality worldwide, which is a serious threat to human life and health. Inflammatory and immune responses are initiated immediately after MI, and unbalanced inflammation post-MI can lead to cardiac dysfunction, scarring, and ventricular remodeling, emphasizing the critical need for an effective inflammation-regulating treatment. With the development of novel therapies, the drug delivery system specific to inflammatory cells offers significant potential. In this review, we introduce immune cells and fibroblasts involved in the development of MI and summarize the newly developed delivery systems related to the use of injectable hydrogels, cardiac patches, nanoparticles, and extracellular vesicles (EVs). Finally, we highlight the recent trends in the use of inflammatory cell-targeting drug delivery systems involving different strategies that facilitate the effective treatment of MI.

## 1. Introduction

Cardiovascular disease (CVD) encompasses a range of conditions that impair the heart and blood vessels, including stroke, congenital heart disease, subclinical atherosclerosis, coronary heart disease, heart failure, venous disease, peripheral artery disease, arrhythmias, myocardial infarction, and valvular disorders, which are characterized by sudden onset and high mortality [1]. Among CVD, myocardial infarction (MI) has emerged as a significant global health issue. This condition arises when the coronary arteries experience a decrease in blood flow for various reasons, leading to a shortage of oxygen in cardiomyocytes and resulting in ischemic necrosis within the affected region [2]. Generally, MI can be categorized into two types based on electrocardiogram (ECG) diagnostic criteria: ST-segment elevation myocardial infarction (STEMI) and non-ST-segment elevation myocardial infarction (NSTEMI) [3]. Clinically, MI is classified into five distinct types: spontaneous myocardial infarction (types 1, 2, and 3) and peri-procedural myocardial infarction (types 4 and 5). These types manifest varying levels of symptoms, with severe instances of MI potentially leading to sudden cardiac arrest [3].

Myocardial infarction (MI) imposes a huge medical and economic burden on society. According to the Report on Cardiovascular Health and Diseases in China 2022, the mortality rates of acute myocardial infarction (AMI) are as high as 78.65 per 100,000 people in rural areas and 60.29 per 100,000 people in urban areas, respectively [4]. The current treatment approaches, involving such treatments as thrombolytic medications, percutaneous coronary intervention (PCI), and coronary artery bypass grafting (CABG), all aim to reestablish blood flow to ischemic cardiomyocytes and demonstrate significant therapeutic benefits [5]. However, these treatments are less effective in patients with severe myocardium infarction and heart failure and are primarily unable to delay the progression of pathological cardiac remodeling; furthermore, they cannot either resolve inflammation response or induce tissue repair. Therefore, it is difficult to reconstruct the damaged myocardium [6,7]. To overcome these obstacles, it is urgent to pursue effective therapeutic strategies for regeneration and restore myocardial function after MI.

Typically, pathological changes in myocardial infarction are gradual processes, which comprise three primary stages, namely inflammation, proliferation, and remodeling [8]. Inflammation is considered a key pathogenic driver of MI, and excessive and chronic inflammation post-MI eventually leads to heart failure [9]. Hence, understanding the key mechanisms related to inflammation will facilitate further treatment development. Furthermore, more novel delivery systems, including hydrogels, nanoparticles, and cardiac patches, have been reported in the last few decades [10]. In virtue of the progress of research into delivery systems, it is achievable to develop a targeted drug delivery system offering a promising strategy for the precise administration of therapeutic agents to damaged heart tissues. Consequently, there is a need in MI therapy to create more targeted treatments that are more effectively designed to influence inflammatory processes.

In particular, this review briefly introduced the immune cells and fibroblasts that are specific on the stage of inflammation. In addition, physicochemical properties and the development of advanced delivery platforms have been presented, which have potential applications in preventing and treating MI. Finally, we summarize the latest strategies, such as injectable hydrogels, nanoparticles, cardiac patches, and extracellular vesicles, involved in inflammatory-targeted drug delivery for the treatment of MI, mainly from 2020–2024.

## 2. Myocardial Infarction and Inflammation

MI initiates a complex inflammatory cascade that is crucial for acute injury and post-infarction repair. After MI occurs, due to the reduction in the oxygenated blood supply and ischemic injury, the myocardium releases damage-associated molecular patterns (DAMPs), leading to extensive cell death and extracellular matrix (ECM) degradation [11,12,13], as well as inducing an accumulation of reactive oxygen species (ROS) that exacerbate tissue damage [14]. The death cells and matrix fragments recruit neutrophils and monocytes/macrophages through DAMPs, which further release a cascade of inflammatory cytokines, such as tumor necrosis factor-α (TNF-α), interleukin (IL)-1β, and IL-6, resulting in a harmful inflammatory microenvironment that severely impairs the cardiac recovery [15,16]. More recently, in addition to the rapid recruitment of innate immune cells, it has been acknowledged that adaptive immune cells, like T cells and B cells, play a crucial role in the regulation of cardiac repair results after a myocardial infarction. Moreover, depending on which heart region is being investigated, fibroblasts make up approximately 15–25% of the total population of cells in the heart, while immune cells make up around 5–10% [17]. Analogous to macrophages, these fibroblasts adopt a pro-inflammatory phenotype soon after MI, after which they differentiate into myofibroblasts, which secrete anti-inflammatory factors and extracellular matrix proteins to repair and stabilize cardiac tissue [18].

To sum up, cardiac ischemic injury leads to an increased release of inflammatory cytokines, prompting a significant recruitment of neutrophils, monocytes, macrophages, and lymphocytes from the bloodstream. This process disrupts cardiac immune homeostasis and influences the progression of myocardial infarction (MI). It has increasingly been agreed that focusing on these immune cells to regain cardiac balance is crucial for mitigating myocardial injury and expediting cardiac repair, ultimately helping to restore cardiac function. Herein, we provide an overview of the multifaceted roles of cells involved in inflammation in MI in this section, which is imperative for devising more efficacious treatment approaches and fostering more effective cardiac rehabilitation.

### 2.1. The Neutrophils

Neutrophils, a component of the immune system, are recognized as the first leukocytes to be recruited to the inflammatory site in order to fight against inflammation [19] (Figure 1). In myocardial infarction (MI), following the release of DAMPs by necrotic cells and chemokines from resident inflammatory and endothelial cells, neutrophils are immediately recruited into the damage site [20]. Neutrophils could promote tissue repair by scavenging cellular debris and possible pathogens, stimulating angiogenesis, and promoting the lysis of ECM [21]. Nevertheless, the uncontrolled activation and entry of neutrophils result in excessive inflammation within the myocardium. Additionally, increased infiltration of older neutrophils into the heart plays a role in cardiac hypertrophy, fibrosis, and dysfunction [22]. Consequently, preventing neutrophil recruitment and the aging of neutrophils has become a significant therapeutic objective.

Neutrophils can be divided into two phenotypes, N1 and N2, according to their profiles of cytokine production, their capability to activate macrophages, and the surface molecules they express [23]. Neutrophils of the N1 subtype exhibit elevated levels of pro-inflammatory markers, including IL-1β, IL-12a, and TNF-α, similar to classically activated macrophages. In contrast, N2 cells demonstrate a strong expression of anti-inflammatory characteristics, such as IL-10 [23]. A study showed that N2 neutrophils gradually accumulate in the infarcted myocardium due to the polarization of neutrophils [24]. Another study discovered that the polarization of N1 neutrophils increased the size of the cardiac infarct and worsened cardiac dysfunction during the initial phase of myocardial ischemia/reperfusion (I/R) [25]. The results indicate that targeting neutrophil polarization could be a potential therapeutic approach for addressing inflammation and damage induced by myocardial conditions.

Moreover, activated neutrophils secrete significant quantities of reactive oxygen species (ROS), proteolytic enzymes, and neutrophil extracellular traps (NETs), which contribute to an ongoing pro-inflammatory setting [26]. Neutrophil extracellular traps (NETs) are structures formed from chromatin, histones, and protein hydrolases that appear in the vascular system during both infectious and non-infectious diseases [27]. The buildup of NETs resulting from hindered degradation seems to negatively impact the ischemic heart tissue, leading to microvascular blockage, inflammation, and the death of cardiomyocytes [28]. In parallel, the protection against myocardial ischemia/reperfusion injury has been demonstrated through the inhibition of NET formation due to the absence of PAD4 or the degradation of NETs via DNase I, as evidenced by reduced infarct size, decreased infiltration of neutrophils, and enhanced cardiac performance [29]. Therefore, altering the intensity of NET formation could serve as a viable approach for treating myocardial infarction.

In conclusion, neutrophils participate in modulating inflammation by a delicate interaction between an array of cytokines and mediators at different stages [30]. Hence, various treatment strategies targeting neutrophils could involve the activation of neutrophils, inhibition of neutrophils recruitment, polarization toward anti-inflammatory phenotypes, and regulation of the formation of NETs.

### 2.2. The Monocytes and Macrophages

In general, monocytes can be subdivided into classical CD14^++^ CD16^−^ and non-classical CD14^+^ CD16^++^ monocytes in humans, with their respective murine counterparts being Ly6C^high^ and Ly6C^low^ monocytes [31]. In response to MI, circulating monocytes quickly infiltrate into the ischemic myocardium and differentiate into macrophages that replace tissue-resident macrophages [32]. The procedure consists of two stages: first, pro-inflammatory Ly6C^high^ monocytes transform into M1 macrophages, and, subsequently, anti-inflammatory Ly6C^low^ monocytes change into M2 macrophages [33,34] (Figure 1). Macrophages play a crucial role in cardiac homeostasis and myocardial repair, and represent one of the earliest and most dominant subpopulations of cardiac-infiltrating cells in the early phase of MI. Increasing evidence suggests that the prolonged and excessive presence of M1 macrophages aggravates the inflammatory reaction and disrupts the repair phase, whereas the M2 macrophages initiate a reparative healing process by releasing anti-inflammatory cytokines [35]. The pro-healing process triggered by M2 macrophages involves resolving inflammation and promoting tissue repair, which eventually results in decreased adverse remodeling and tissue damage [36]. In addition, M1 macrophages exhibit a notably high expression of specific surface markers, including cluster of differentiation (CD) 80 and CD86. These markers are primarily associated with the inflammatory response, which plays a crucial role in the immune system’s ability to combat pathogens and manage tissue damage. On the other hand, M2 macrophages are distinguished by their elevated expression of a different set of markers, such as CD163, CD206, Arg1, FIZZ1, and YM1. These markers are linked to anti-inflammatory processes and tissue repair, highlighting the contrasting functional roles of these two macrophage subsets in the immune response [37]. These reports underscore the important role of the timely polarization of the inflammatory macrophage phenotype after MI.

The traditional perspective mentioned earlier suggests that macrophages originate from monocytes present in circulation and are categorized as M1 or M2 types. However, it is now understood that macrophages exhibit diverse developmental and tissue-specific functional profiles [38]. For instance, the expression of CCR2 has been employed to classify the phenotype of cardiac macrophages and to differentiate the origins of these cells. CCR2^−^ macrophages originate from the embryonic yolk sac, and fetal liver monocytes persist without the need for monocyte recruitment, in contrast to CCR2^+^ macrophages, which rely on monocyte recruitment for their maintenance [39]. CCR2^−^ macrophages inhibit monocyte recruitment, whereas CCR2^+^ macrophages promote the recruitment of neutrophils and monocytes [40]. Meanwhile, CCR2^+^ macrophages express higher levels of inflammatory chemokines and cytokines, such as monocyte chemoattractant protein1 (MCP-1), IL-1β and TNF-α, which aggravates inflammation following MI [41]. Therefore, depleting the CCR2^+^ macrophages is an alternative strategy for MI treatment.

In summary, the multifaceted roles of macrophages are closely related to their phenotype and secreted cytokines. Therefore, the effective treatments targeting macrophages might include regulating the monocyte/macrophage infiltrate, or/and increasing the M2/M1 ratio, either by inhibiting M1 macrophage polarization or by promoting M1 toward M2 macrophage polarization, which benefits heart function recovery and become the next milestone for myocardial infarction treatment.

### 2.3. The Lymphocytes

In addition to the innate immune cells mentioned previously, cells linked to adaptive immunity also contribute significantly to the inflammatory process that occurs after MI. The primary cellular elements of adaptive immunity are T cells and B cells, which originate from lymphoid progenitors found in the bone marrow [42].

T cells represent a crucial component of the immune system involved in the acquired immune response. These cells can be categorized into αβT cells and γδT cells based on their unique T cell receptors (TCRs). Furthermore, αβT cells can be further classified into CD4^+^ T cells and CD8^+^ T cells [43]. After myocardial injury, CD4^+^ T cells are recruited in the infarcted area and regulate the local innate immune response, especially monocyte infiltration and polarization, which promote heart repair [44]. Following effective activation, CD4^+^ T cells have the ability to differentiate into different subsets, which include effector T cell types, like Th1, Th2, Th3, Th9, Th17, and Th22, as well as T regulatory (Treg) cells, T follicular helper (Tfh) cells, and T follicular regulatory (Tfr) cells. This differentiation is primarily influenced by the cytokines they secrete and the expression of surface markers [45]. These different T cell subsets have diverse expression patterns facilitating immune response (Figure 1). Tregs represent a specific subset of T cells, characterized by CD4^+^CD25^+^Foxp3^+^ markers [46]. After the MI, Treg cells infiltrate into the damaged heart and protect the impaired myocardium by means of increasing collagen content and promoting scar formation in the infarction area [47]. Furthermore, Tregs produce anti-inflammatory cytokines, including TGF-β and IL-10, which have strong effects in maintaining immune homeostasis [48]. At the same time, Treg cells hinder the gathering of inflammatory cells and reduce the localized production of pro-inflammatory cytokines [49]. Hence, Treg cells appear to inhibit inflammation and participate in myocardial repair after MI. Other CD4^+^ T cells phenotypes, like Th1, Th9, and Th17 cells enhance inflammation by increasing neutrophil infiltration, which exacerbates myocardial damage, whereas Th2 and Th22 cells suppress excessive inflammatory responses by inhibiting leukocyte accumulation, which improves cardiac function post-MI [50]. A study found an increase in Th17 cells and a decrease in Treg cells in MI, resulting in the overproduction of pro-inflammatory cytokines, such as IL-17A, IL-17F, and IL-22 [51]. Thereby, adjusting the phenotype of T cells and managing the Treg/Th17 ratio may result in better regulation of the immune response, ultimately contributing to the advancement of more effective myocardial infarction therapies.

CD8^+^ T cells play a role in immune response; however, their involvement in inflammatory response after MI remains unclear. Recent research has unraveled the deleterious role of CD8^+^ T cells after acute ischemia. Research has indicated that a reduction in CD8^+^ T cells can lead to a decrease in apoptosis, hinder the inflammatory response, restrict myocardial damage, and improve cardiac function [52]. These findings suggest that targeting harmful CD8^+^ T lymphocytes may offer new therapeutic approaches for treating acute myocardial infarction. However, another study showed that CD8^+^ T cells have both a beneficial and detrimental role on account of the potential of CD8^+^ T cells to induce cardiac damage initially, although they may promote a healing effect later on because of the expression of AT2R [53]. Further investigation is essential to achieve a thorough understanding of CD8^+^ T cells and their role in the process of MI.

B lymphocytes have been demonstrated to play a central role in the activation of the inflammatory cascade. After MI, the entry of mature B cells into the damaged region of the myocardium triggers pro-inflammatory monocytes to enhance the inflammatory response, which subsequently reduces myocardial contractility, encourages apoptosis, and aggravates left heart function [54]. On the other hand, the accumulation of B cells following MI leads to a preferential production of IL-10. This cytokine aids in diminishing inflammation, mitigating myocardial damage, and safeguarding cardiac function [55].

On the whole, the complex roles of T cells and B cells after MI have been established. Effective management of immune cells in a coordinated manner might result in improved regulation of the overall immune response.

### 2.4. The Fibroblasts

In addition to aforementioned immune cells, the fibroblasts/myofibroblasts are also strongly related to inflammation regulation in MI. Cardiac fibroblasts (CFs) are mesenchymal cells that are derived from the embryonic mesoderm, which undergo epithelial-to-mesenchymal (EMT) and endothelial-to-mesenchymal (EndMT) transitions to migrate into the heart early in development [56]. CFs are the primary cells responsible for the synthesis of ECM and play a crucial role in maintaining normal heart function [57]. CFs display unique phenotypes at various time intervals following myocardial infarction (MI). During the early phase, prompted by the significant loss of cardiomyocytes in the affected region, the resting cardiac fibroblasts release pro-inflammatory cytokines, such as interleukin (IL)-1β and IL-6, which play a role in promoting inflammation. In the subsequent days, influenced by an array of growth factors, including TGF-β, FGF-2, and PDGF, CFs gradually transdifferentiate into myofibroblasts and generate mediators that are both anti-inflammatory and proangiogenic, thereby aiding in the development of granulation tissue [58].

The transformation of fibroblasts into myofibroblasts is essential in mediating the inflammatory response and facilitating repair processes following myocardial infarction (MI). The latest study showed that ALKBH5 could induce fibroblast-to-myofibroblast transformation [59]. Activated fibroblasts express fibroblast activation protein alpha (FAPα), dipeptidyl peptidase IV, and MMPs, which break down the ECM and collagen I in the damaged area and clear damaged cells and tissue debris, as well as promoting growth by releasing growth factors bound to the ECM [60]. Research has shown that activated fibroblasts can phagocytose dead cells through secrete the milk fat globule-epidermal growth factor 8 (MFG-E8), which promotes apoptotic engulfment as well as reducing inflammatory responses [61].

In brief, promoting the differentiation of fibroblasts toward myofibroblasts can serve as a regulatory tactic to mitigate the inflammatory response.

## 3. Delivery Systems

Drug delivery systems (DDSs) were originally devised to enhance the bioavailability of drugs and reduce dosage frequency, but in recent years they have been repurposed to facilitate new therapeutic modalities [62]. Targeted drug delivery systems are preferred over conventional drug delivery systems and have emerged as a potent strategy for maximizing therapeutic benefits while lowering the risk of negative effects. For instance, injectable functional hydrogels were designed in tandem with drug delivery systems to target inflammatory issues in infarcted areas for MI treatment, which maximizes drug utilization, and increase tolerance to potential injection errors [63]. Nanoparticles designed for the transport of drug formulations can additionally focus on the injured heart, with validation through animal models demonstrating significant promise for MI treatment [64]. Concurrently, cardiac patches and extracellular vesicles have led to extensive research in targeted delivery; they can load and deliver cargo to the damaged region, reducing nontargeted side effects [65,66]. These delivery systems are opening new avenues for MI therapy.

### 3.1. Hydrogels

Hydrogels are well known as highly hydrated three-dimensional (3D) network structure, which offer significant potential in the medical and biomedical areas of MI therapy as effective drug-delivery systems, due to their remarkable biocompatibility, ability to undergo chemical modifications, and adjustable physical properties, in addition to relatively simple processing methods [67]. Generally, hydrogels can be categorized into three main groups: natural biomaterial hydrogels, synthetic biomaterial hydrogels, and smart response-based hybridized hydrogels (Figure 2A) [68]. Natural hydrogel refers to hydrogels prepared from natural biological materials (i.e., alginate, agarose, collagen, gelatin, chitosan, hyaluronic acid, fibronectin, pectin, keratin, ECM, etc.) [69]. Synthetic hydrogels, on the other hand, are 3D polymeric structures (i.e., polyvinyl alcohol, polyacrylamide, polyethylene glycol hydrogels, etc.) that are artificially created to possess specific shapes and mechanical properties [70]. Furthermore, smart response-based hybrid hydrogels are a special type of hydrogel that combines the properties of multiple materials to achieve sensitive responses to environmental changes [68]. It is worth mentioning that several smart hydrogels have been developed to react to specific stimuli, such as physical (temperature, light, electromagnetic fields, pressure, and ultrasound radiation), chemical (pH, glucose, and ionic strength), or biological (enzymes and antigens/antibodies) stimuli, in order to meet the intricate requirements of specific conditions [71].

Furthermore, hydrogels can possess a range of versatile properties by adjusting their chemical compositions, utilizing different crosslinking strategies, and modifying their physical structures [72]. For instance, injectable hydrogels can be created through chemical or physical crosslinking to enable injectability, which guarantees a controlled and sustained release of the drug at the designated location [73]. Moreover, injectable hydrogels could realize the minimal invasiveness of lesions and substantially diminish the negative effects linked to systemic medication exposure, resulting in increased treatment effectiveness and better patient comfort and adherence [74]. The utilization of injectable hydrogels has demonstrated substantial potential in facilitating the overall healing process following MI.

### 3.2. Nanoparticles

Nanoparticles are typically characterized as particles that fall within a size range from 1 to 100 nanometers. They can be categorized into two primary types: organic and inorganic nanoparticles. Organic nanoparticles include a variety of forms, such as polymer nanoparticles, dendrimers, polymeric micelles, nanogels, liposomes, and solid lipid nanoparticles (Figure 2B). On the other hand, inorganic nanoparticles encompass other materials, such as metal nanoparticles, silica nanoparticles, and quantum dots (Figure 2B). This classification underscores the diverse nature of nanoparticles and their significant roles in various applications across multiple fields, such as medicine, electronics, and environmental science [75,76]. Nanomaterials are produced at the nanoscale and exhibit a longer circulation duration because of their capacity to evade clearance by the kidneys and capture by the reticuloendothelial system following surface modifications. Furthermore, these nanomaterials can be adjusted to possess controllable physical or chemical attributes, which helps enhance their drug pharmacokinetics, enabling active targeting or various specific functions [77]. Given the aforementioned advantages of nanoparticles, they are anticipated to significantly advance MI diagnosis and treatment. Among the various nanoparticles, liposomes stand out as the sole nanocarriers that have received approval from the US Food and Drug Administration (FDA) for use in clinical settings [78]. Liposomes utilized as delivery systems for drugs exhibit notable benefits, such as outstanding biodegradability, minimal immunogenic response, the increased biocompatibility of the drug, fewer side effects, and augmented therapeutic effectiveness [79]. In addition, polymer nanoparticles, polymeric micelle, and silica nanoparticles have also contributed to huge progress in MI treatment. Polymer nanoparticles are crafted from diverse natural or synthetic polymeric sources, which can obtain specific properties with different polymer materials. Biodegradable polymers find extensive use in the creation of effective delivery systems that offer beneficial characteristics, including biodegradability, compatibility with biological systems, reduced toxicity, satisfactory stability, natural renewability, and affordability [80]. Micelles consist of colloidal dispersions made up of amphiphilic substances, typically employed for the administration of hydrophobic medications, and range in size from 10 to 100 nm [81]. Micelles exhibit an effective core–shell configuration, dynamic stability, and enhanced solubility for hydrophobic pharmaceuticals [82]. Silica nanoparticles are a type of crystalline particle that consists of silica with an orderly arrangement. The siloxane framework and silanol groups in these nanoparticles facilitate bonding with various biomolecules to attain the intended objective [83]. Examples related to nanoparticles targeted for myocardial infarction treatment are introduced in Section 3.2.

### 3.3. Cardiac Patches

Cardiac patches are composed of stents or loaded therapeutic agents (cells or bioactive molecules) that provide mechanical reinforcement, synchronous electrical conduction, and local delivery within the infarcted area to promote cardiac recovery [84]. Commonly, based on the source of synthetical biomaterials, cardiac patches can be categorized into two main types, namely natural and synthetic patches [85]. Natural cardiac patches can be synthesized from in vivo sources, like proteins, polysaccharides, decellularized extracellular cardiac matrix (dECM), and even cell sheets, whereas synthetic cardiac patches can be fabricated with a great variety of polymers [85]. However, a more nuanced classification can be implemented according to the property. The most promising types of patch are classified into five types, including hydrogel patches, fibrous films, elastomer patches, cell sheets, and microneedle patches (Figure 2C) [65]. These unique types of cardiac patches possess specific features and advantages, offering a wide range of potential applications.

### 3.4. Extracellular Vesicles (EVs)

Researchers have shown significant interest in extracellular vesicles (EVs) as potential natural systems for drug delivery. According to the minimum information for studies of extracellular vesicles (MISEV) 2023, extracellular vesicles is the term for particles that are delimited by a lipid bilayer and cannot replicate on their own (a vesicular component of extracellular particles) [86]. Based on their size and origin, extracellular vesicles are mainly divided into exosomes, microvesicles, and apoptotic bodies (Figure 2D) [87]. Exosomes are nanoscale vesicles, typically measuring 30 to 150 nanometers (nm) in diameter, and are generated within endosomes, which are cellular compartments involved in the internalization and degradation of cellular components. Microvesicles, on the other hand, are larger, ranging from 100 to 1000 nm in size, and are produced by the budding and detachment of pieces of the cell membrane, a process that can be facilitated by membrane protrusions. Apoptotic bodies are the largest of these vesicles, exceeding 1000 nm in diameter, and they are released as a byproduct of apoptosis (programmed cell death).

In particular, EVs can carry a complex of cargo, including proteins, lipids, and nucleic acids, which is selectively packaged and delivered to target cells [88]. Once the cargo enters the cell, it can regulate the physiological state of the target cells. For example, EVs have been found could carry miR-129 to contribute to ameliorating inflammation, thus mitigating myocardial I/R injury [89]. EVs can be established as therapeutic systems and have led to extensive research into targeted drug delivery because of their high physicochemical stability, which can protect therapeutic agents from endogenous enzymatic degradation, as well as their low immunogenicity and high biocompatibility [90]. In addition, EVs are able to penetrate tissues directly and communicate with other cells over long distances, interact with specific molecules, participate in endocytosis, and fuse with the cell membrane [91]. Furthermore, modifications of EVs enable researchers to improve the specificity of drug delivery, reduce off-target effects, and minimize systemic toxicity [92].

## 4. Inflammatory Cells Targeted Therapy Strategies

As previously mentioned, the immune cells, including neutrophils, macrophages, lymphocytes, and cardiac fibroblasts, play a pivotal role in the inflammatory response after MI. With an improved understanding of the complex microenvironment in the MI area, researchers have actively developed targeting delivery systems, such as injectable hydrogels, biomimetic nanoparticles, cardiac patches, and extracellular vesicles. Hence, the following sections will focus on the current studies on using various delivery technologies to target the pathological aspects of inflammation in MI to achieve inflammatory cell-targeted modulation for MI treatment, primarily summarizing the recent research from 2020–2024.

### 4.1. Targeting Neutrophils

Neutrophils, as the first wave of leukocytes to infiltrate the injured myocardium, maintain the balance of inflammation. In recent research, it has been shown that the polarization of neutrophils plays a significant role in both the inflammation process and its resolution. Therefore, managing the recruitment of neutrophils and modifying their phenotypes represent key therapeutic targets in addressing inflammation and injury caused by myocardial infarction (MI).

Reports have highlighted a range of functional injectable hydrogels that enhance MI repair by modulating the infiltration of neutrophils. In a study, Wang et al. presented a novel exosome-loaded conductive hydrogel designed to enhance myocardial repair by using adipose-derived stem cell (ADSC) exosomes encapsulated within a hyaluronic acid-dopamine (HA-DA) hydrogel and incorporated black phosphorus (BP) into the hydrogel, which concurrently improved the biological and electrical signals. In vivo experiments found that Gel-BP@PDA-Exo attenuated the local inflammatory response, leading to reduced neutrophil infiltration [93].

Nanoparticles are capable of encapsulating and transporting drug molecules with enhanced delivery efficiency and therapeutic efficacy. AI-Darraji et al. investigated a liposomal azithromycin (AZM) formulation (L-AZM) in a clinically relevant study and found a remarkable decrease in cardiac inflammatory neutrophils as well as the infiltration of inflammatory monocytes. The immunomodulatory effects of L-AZM were associated with a reduction in cardiac cell death and scar size as well as enhanced angiogenesis [94]. Chen et al. fabricated biomimetic liposomes (Neu-LPs) by fusing neutrophil membranes with conventional liposomes that inherited the surface antigens of the source cells, making them ideal decoys of neutrophil-targeted biological molecules. Consequently, Neu-LPs demonstrated substantial therapeutic effectiveness by reducing neutrophil infiltration in the area surrounding the infarct, thereby offering cardiac protection, mitigating severe inflammation, and enhancing angiogenesis [95]. Shao et al. developed a nonviral delivery system for nucleic acids that includes an EC-specific polycation (CPC), which is made from ethanolamine-modified poly (glycidyl methacrylate) grafted with CRPPR. This system is capable of effectively co-delivering siR-ICAM1 along with pCXCL12 for treating MI. The therapeutic combination of CPC/siR-ICAM1 alongside CPC/pCXCL12 led to a significant reduction in Ly6G^+^ neutrophils and F4/80^+^ macrophages, thereby effectively reducing the inflammatory response, enhancing angiogenesis, and improving cardiac function without any reported side effects [96]. Han et al. effectively created a biomimetic IL-5 nanoparticle by encapsulating it within a neutrophil-derived membrane. The introduction of NM-NPIL-5 nanoparticles has the potential to decrease neutrophil infiltration while promoting M2-like polarization of macrophages in the aftermath of acute myocardial infarction (AMI) [97]. Hou et al. created nanocomplexes (NCs) that target endothelial cells and are ultra-sensitive to ROS to facilitate the effective co-delivery of VCAM-1 siRNA (siVCAM-1) alongside dexamethasone (DXM). This combined approach effectively hindered the recruitment of neutrophils by disrupting both their migration and adhesion [98].

In addition, Yuan et al. developed a biocompatible microneedle patch to deliver exosomes containing an miR-29b mimic for MI treatment. In a mouse model of MI, implantation of this composite patch could alleviate neutrophil infiltration and decrease F4/80 positive macrophages, facilitating cardiac repair post-MI [99]. Feng et al. demonstrated that MSC-derived exosomes packaged with miR-199 treatment could attenuate myocardial I/R injury by reducing neutrophil infiltration and NETs formation within the heart [100].

Prompt removal of activated neutrophils from the injured heart minimizes inflammation caused by neutrophils and ultimately protects cardiac function. Kim et al. described a poly (lactic-co-glycolic acid) (PLGA) nanoparticle that provides spatiotemporal control over neutrophil fate to both stymie MI pathogenesis and promote healing. They created PLGA nanoparticles (RC NPs) loaded with roscovitine and catalase that respond to hydrogen peroxide (H_2_O_2_). Furthermore, they exhibited that administering a single intravenous dose of RC NPs in myocardial infarction (MI) rats leads to significant apoptosis of neutrophils in the infarcted heart [101].

In addition, Hong et al. developed a photocrosslinkable gelatinmethacryloyl (GelMA) hydrogel for delivering human vascular progenitor cells utilizing an in situ photopolymerization approach. After the hydrogel is introduced into the damaged areas of mouse hearts, the vascular cells that integrate polarize the bone marrow-derived neutrophils towards a non-inflammatory N2 phenotype through transforming growth factor beta (TGF-β) signaling, thereby promoting a pro-regenerative microenvironment [102].

The above study emphasizes the significance of manipulating neutrophils in a multi-dimensional manner for anti-inflammatory therapies aimed at restoring cardiac function.

### 4.2. Targeting Monocytes and Macrophages

In recent years, monocytes/macrophages have become increasingly attractive as potential targets for improving myocardial repair. The monocyte/macrophage-targeted treatment strategies can be broadly categorized into two distinct types, which include regulating their infiltration and modulating their phenotypes.

#### 4.2.1. Targeting Monocytes

It is well known that the excessive arrival of monocytes would perpetuate inflammation at the site of injury. Several delivery systems which target the infarcted heart to reduce the recruitment of monocytes have been developed. For instance, Zhu et al. developed a novel injectable electroconductive hydrogel by incorporating irbesartan as a drug delivery system for cardiac repair. The P-DNAH-I hydrogel that was developed exhibited remarkable biocompatibility along with a swift initial release of irbesartan. When injected into hearts affected by infarction in MIRI mouse models, this hydrogel successfully prevented the recruitment of Ly6C^high^ CCR2^+^ inflammatory monocytes from peripheral blood to the heart, thereby diminishing the inflammatory response and decreasing the size of the infarction [103]. In 2021, Ikeda and colleagues created nanoparticles of poly-lactic/glycolic acid (PLGA) that incorporated either cyclosporine A or pitavastatin to reduce the recruitment of Ly6C^high^ monocytes to the MI site, leading to a reduction in inflammation and an improvement in the size of the infarct following MI [104]. Qiao et al. demonstrated that Krüppel-like Factor 2 (KLF2)-EVs attenuated myocardial injury in mice via shuttling miR-24-3p that restrained the Ly6C^high^ monocytes recruitment from bone marrow by inhibiting CCR2 expression [105].

In addition, delivery systems that directly interact with monocytes have been reported. Richart et al. demonstrated that Apo AI nanoparticles directly interact with neutrophils and monocytes partially via SR-BI and reduce monocyte activity, with a reduction in the number of circulating leukocytes after n-apo AI infusion [106]. Weng et al. created a bionic platelet platform for the delivery of RvD1 that responds to ROS. This innovative formulation facilitates the directed transport of RvD1 to the site of injury by utilizing circulating chemotactic monocytes, meanwhile allowing for localized release and ultimately leading to significant improvements in cardiac function [107]. Li et al. developed a mesoporous silica nanoparticle (MSN-NGR1-CD11b) that is conjugated with the CD11b antibody and loaded with Notoginsenoside R1 (NGR1). This formulation enabled precise targeting of CD11b-expressing monocytes and neutrophils in a noninvasive way. The intravenous injection of the MSN-NGR1-CD11b antibody nanoparticle into mice with MI significantly enhanced cardiac function and promoted angiogenesis by increasing the targeting of the damaged myocardium [108]. Tan et al. developed platelet-like fusogenic liposomes (PLPs). When coated with these PLPs, mesoporous silica nanospheres carrying miR-21, known for its anti-inflammatory properties, are capable of specifically targeting and delivering to inflammatory monocytes within the bloodstream of mice subjected to MI/R. Subsequently, the nanospheres enter the cytoplasm of monocytes via membrane fusion, facilitating the reparative reprogramming of the inflamed macrophages derived from them. Administration of this formulation in vivo can significantly maintain cardiac function in mice experiencing MI/R [109].

#### 4.2.2. Targeting Macrophages

Macrophages secrete and released a large number of inflammatory factors, which contribute to the exacerbation of myocardial injury. Hence, the delivery systems targeting macrophages to regulate inflammation are the most prevalent type.

First of all, a significant portion of the research has primarily focused on reducing the infiltration of macrophages into the infarcted area. Rocker et al. created a thermo-responsive injectable gel that is made from chitosan and poly (*N*-isopropylacrylamide), along with sulfonate groups, aimed at facilitating the spatiotemporal delivery of proteins to safeguard cardiac function following MI. This innovative delivery system demonstrated that the regulated release of vascular endothelial growth factor (VEGF), IL-10, and platelet-derived growth factor (PDGF) decreased macrophage infiltration while enhancing vascularization [110]. Similarly, an innovative magnetic vagus nerve stimulation (mVNS) system was described by Sun et al., which featured an injectable hydrogel made from chitosan/β-glycerophosphate (CS/GP) that contained superparamagnetic iron oxide (SPIO) nanoparticles, together with a gentle magnetic pulse sequence. This injectable hydrogel, formulated from materials which are safe for clinical use, facilitated a minimally invasive implantation process, while the SPIO nanoparticles embedded in the hydrogel enabled precise magnetic activation of a singular vagus nerve. Analysis through immunohistochemistry on myocardial tissue revealed that the mVNS treatment was effective in reducing macrophage infiltration and lowering the levels of inflammatory factors (such as IL-1β and TNF-α) in the plasma post-myocardial infarction. Collectively, these findings indicated that mVNS holds considerable promise for clinical applications in the treatment of MI [111]. Zhang et al. developed an injectable conductive hydrogel by attaching polypyrrole (PPy) to non-conductive gelatin and integrating these components into a gel system created through the Schiff base reaction between oxidized xanthan gum (OXG) and gelatin. Upon the injection of the self-healing OGGP3 (3 wt% GP) hydrogel into the myocardial scar tissue of rats, it demonstrated anti-inflammatory properties, evidenced by a reduction in CD68^+^, enhanced cardiac function, diminished myocardial fibrous tissue, and an increased conduction velocity within the myocardial tissue [112]. Liu et al. developed an injectable hydrogel through the combination of natural alginate and Au@Pt nanoparticles (Au@Pt/Alghydrogel) to encapsulate brown adipose stem cells (BASCs). The effectiveness of the Au@Pt/Alg hydrogel containing BASCs has been assessed, demonstrating a reduction in inflammatory factors associated with macrophages. These delivery systems are particularly skilled at altering the microenvironment, which plays a vital role in the treatment of MI [113].

Apart from injectable hydrogels, nanoparticle-mediated blocking of excessive infiltration of macrophages in infarcted myocardium could, likewise, be an effective intervention approach. Guan et al. developed PCNP/O_2_ and explored the effects of injecting oxygen-releasing nanoparticles on the inflammatory response within infarcted heart tissue. The results from immunofluorescence staining indicated a significant decrease in the density of CD68 positive pro-inflammatory macrophages following the administration of PCNP/O_2_ nanoparticles, which suggests that the introduced oxygen-releasing nanoparticles helped reduce tissue inflammation in the damaged heart [114].

In addition, Bannerman et al. synthesized a cardiac patch using a novel material, poly (itaconate-co-citrate-co-octanediol) (PICO), encapsulating citric acid (CA) and itaconic acid (ITA) components. The bio-fabricated PICO patches release CA and ITA upon degradation in accelerated and hydrolytic conditions. Furthermore, PICO patches resulted in less macrophage infiltration and a reduced inflammatory response, demonstrating cardioprotective effects on cardiac cells after ischemic injury [115]. Milano et al. reported that cardiac progenitor cell-derived exosomes highly enriched in miR-146a-5p significantly reduced CD68^+^ macrophages infiltrates into the heart [116]. Mentkowsk et al. engineered the surface of CDC-derived EVs to express a CM-specific binding peptide (CMP) and characterized for size, morphology, and protein expression. The surface-engineered CMP- EVs reduced CD68^+^ macrophage infiltration post-MI, and improved LVEF; they also reduced cardiomyocyte apoptosis, demonstrating a strategy to optimize therapeutic EV delivery post-MI [117].

Subsequently, a substantial part of the studies primarily concentrated on reprogramming macrophage polarization. A growing body of evidence shows that alternate macrophage phenotypes alter the inflammatory response, resulting in protection against complications related to MI. In conjunction with the synchronized improvement of delivery systems, different strategies by means of hydrogels, nanoparticles, cardiac patches, or EVs provide an alternative strategy for MI treatment.

On the one hand, a great deal of research has centered on inhibiting the polarization of M1 macrophages. According to the latest research, an injectable hydrogel consisting of oxidized hyaluronic acid and polylysine (OHA-PL) was developed to effectively encapsulate exosomes derived from adipose tissue mesenchymal stem cells (ADSC-Exos) and enhance their retention in physiological environments. The OHA-PL@Exo hydrogel demonstrated the ability to eliminate reactive oxygen species, reduce the expression of the M1 macrophage-related protein CD86, and diminish inflammation during the initial stage of myocardial infarction (MI) [118]. Li et al. created a delivery system for microRNA-21-5p by utilizing functionalized mesoporous silica nanoparticles (MSNs) embedded in an injectable hydrogel matrix (Gel@MSN/miR-21-5p). They showed that the MSN complexes released significantly reduced the inflammatory response by preventing the polarization of M1 macrophages in the affected myocardium [119]. Similarly, Feng et al. introduced a novel injectable hydrogel made from puerarin and chitosan, created through in situ self-assembly, which concurrently delivers mesoporous silica nanoparticles (CHP@Si) aimed at aiding myocardial repair following MI. Their findings indicate that puerarin released from the CHP@Si hydrogel influences the inflammatory response by suppressing the M1-type polarization of macrophages and reducing the expression of pro-inflammatory factors [120]. Hu et al. fabricated a novel active targeting biomimetic nanomedicine (At@NQA) using an amphiphilic calixarene molecule (QA) load with the anti-inflammatory and anti-fibrotic atorvastatin calcium (At). The myocardial I/R injury experiments in mice showed that the At@NQA could efficiently target the damaged myocardium for better bioavailability and drug delivery efficacy and inhibit the oxidative damage, thereby inhibiting the M1 polarization of macrophages, alleviating the excessive inflammation, and improving the cardiac functions [121]. A study indicated that exosomes secreted by bone marrow mesenchymal stem cells harboring miR-25-3p (BMSC-Exo-25-3p) could attenuate the inflammatory response by inhibiting M1-like macrophage polarization and pro-inflammatory cytokine expression; the mechanism lies in the inactivation of JAK2/STAT3 signaling pathway [122].

On the other hand, the transition of macrophages from a pro-inflammatory M1 type to an anti-inflammatory or pro-healing M2 type is helpful in reducing chronic inflammation. There is plenty of research on using injectable hydrogels for targeted macrophage polarization. For example, Zhang et al. created an injectable hydrogel that mimics the ECM for encapsulating artificial apoptotic cells (AACs) alongside VEGF. This delivery system, designed with careful control of AACs and VEGF, facilitates precise spatiotemporal modulation of macrophage polarization. This process enhances the rapid transition of M1 macrophages to M2 macrophages, helps to reduce inflammation, and simultaneously promotes vascularization in the affected infarcted area, ultimately leading to notable enhancements in cardiac function following MI [123]. In a comparable way, Chen et al. developed a matrix metalloproteinase (MMP)2/9-responsive hydrogel system (MPGC4) consisting of tetra-poly (ethylene glycol) (PEG) hydrogels and a composite gene nanocarrier (CTL4) that is composed of carbon dots (CDots) coupled with interleukin-4 plasmid DNA via electrostatic interactions. MPGC4 can be automatically triggered to release CTL4 on demand after MI to regulate the infarct immune microenvironment. The in vitro results demonstrate that CTL4 promotes pro-inflammatory M1 macrophage polarization to the anti-inflammatory M2 subtype and contributes to cardiomyocyte survival through macrophage transition [124]. In addition, Luo et al. developed a new type of puerarin hydrogel-SDF-1α@PUE, which can enhance the homing of endogenous stem cells through intrapericardial injection, while polarizing the recruited monocytes/macrophages into the M2 phenotype, reducing inflammation, protecting myocardial cells, promoting angiogenesis, improving cardiac function, and reducing the risk of arrhythmia [125]. A pH/ROS dual-responsive injectable hydrogel encapsulating polydopamine–rosmarinic acid nanoparticles and adding conductive composites was developed by the same group. The OGDPR hydrogel, which is conductive and loaded with multiple drugs, exhibited anti-inflammatory properties and facilitated the polarization of macrophages towards the M2 phenotype in vivo. This action created a more supportive microenvironment for myocardial cells within the infarct zone, decreased cell apoptosis, slowed the decline of functional cardiomyocytes in the affected area, and contributed positively to the preservation of cardiac structure and the protection of cardiac function, ultimately aiding in the management of MI [126]. Liu et al. developed an injectable carrier that could respond to the inflammatory microenvironment in the early stage of MI with the rapid release of curcumin (Cur), followed by a sustained release of recombinant humanized collagen type III (rhCol III) to treat MI. The prepared GelB-Cur NPs and rhCol III-loaded multifunctional carrier could inhibit the polarization of RAW264.7 cells to the M1 phenotype and promote differentiation to M2 macrophages [127]. Another investigation demonstrated that the engineered hydrogel platform TPL@PLGA@F127 could direct macrophages to adopt an anti-inflammatory M2 macrophage phenotype within the initial 3 days. This transition could enhance the reduction in the inflammatory process in MI and better safeguard the cardiomyocytes during the early phase [128]. MNPs/Alg hydrogel, composed of natural substances from two marine sources, downregulates pro-inflammatory M1 macrophages and induces macrophage phenotypic transformation to anti-inflammatory M2 type 1–3 days after treatment. Therefore, this composite hydrogel exhibits a synergistic effect in modulating the early inflammatory microenvironment of myocardial infarction and facilitating macrophage transformation, thereby alleviating the progression of MI [129]. Cimenci et al. showed that therapy utilizing collagen type I hydrogel loaded with fisetin (fisetin-HG) can diminish the accumulation of methylglyoxal-advanced glycation end-products (MG-AGEs) and lower oxidative stress in the MI heart, correlating with a reduction in scar size and the enhancement of cardiac function. Additionally, fisetin-HG treatment encourages neovascularization and raises the quantity of pro-healing macrophages within the infarct region, concurrently decreasing the presence of pro-inflammatory macrophages [130]. Zhang et al. developed a new injectable composite hydrogel (CS-Gel-BP@PDA) by combining chitosan (CS) and gelatin (Gel) with polydopamine-modified black phosphorus nanosheets (BP@PDA), achieving sustained release in the MI area to scavenge excess ROS, alleviate oxidative stress, and inhibit the inflammatory response by promoting the polarization of macrophages toward the M2 phenotype, thereby improved myocardial repair and cardiac function after MI [131]. Lin et al. constructed a four-in-one injectable hydrogel (TA-Arg-Qc-Mn2^+^) through chemical bonding and supramolecular interactions, loading four selected drugs with different functions (thioctic acid, quercetin, arginine, and Mn2^+^). The TA-Arg-Qc-Mn2^+^ hydrogel significantly promoted the expression of CD206 without affecting the expression of CD68. Moreover, the hydrogel extract promoted the secretion of IL-4 and slightly inhibited the secretion of IFN-γ in the macrophages, which consistently confirms that the hydrogel could induce M2-type macrophage polarization. This is important for suppressing the inflammation in the microenvironment of myocardial injury [132]. Tan et al. introduced a bioengineered injectable hyaluronic acid hydrogel designed to optimize the local delivery efficiency of trophoblast stem cells derived-exosomes. This unique exo-hyper gel system is founded on the synergy between hyperbranched poly (β-hydrazide esters) (HB-PBHE) and thiolated hyaluronic acid (SH-HA), tailored for the delivery of exosomes from iPC trophoblast stem cells. The TSC exosomes promoted the transformation of M1 macrophages into M2 macrophages in the infarcted and peripheral zones and regulated the inflammation. Meanwhile, iPC delivery of exo-hyper gel enhanced this effect [133]. An injectable hydrogel exhibiting pH- and temperature-responsive properties for the localized delivery of oncostatin M (OSM) was developed by Jiang et al. The composite OSM@ CS-CA-PNIPAM hydrogel can significantly improve the microenvironment by promoting M2-type polarization of macrophages as well as by inhibiting the secretion of inflammatory cytokines (TNF-α and IL-6), thus ultimately improving cardiac function [134]. Wang et al. introduced a novel supramolecular compound, NapFFY, designed to co-assemble with IL-10 and SN50, forming an innovative anti-inflammatory hydrogel, SN50/IL-10/NapFFY, that exhibits cardioprotective properties. In a rat model of MI, the intramyocardial injection of the SN50/IL-10/NapFFY hydrogel significantly reduced the levels of pro-inflammatory cytokines and enhanced the polarization of M2 macrophages, leading to decreased apoptosis of cardiomyocytes and improved vascularization in the border zones [135]. Zhang et al. formulated an injectable alginate hydrogel incorporated with annexin A1 (AnxA1), which serves as an intrinsic anti-inflammatory and pro-resolving mediator, aimed at treating MI. The findings from in vitro experiments indicated that the composite hydrogel decreased the abundance of pro-inflammatory macrophages while enhancing the proportion of pro-healing macrophages, facilitated by the adenosine monophosphate (AMP)-activated protein kinase (AMPK) and mammalian target of rapamycin (mTOR) signaling pathway. Moreover, administering this composite hydrogel through intramyocardial injection in a mouse model of MI effectively influenced the transition of macrophages towards pro-healing M2 phenotypes. This shift alleviated acute inflammatory responses alongside cardiac fibrosis, stimulated angiogenesis, and led to an improvement in cardiac function [136].

In addition, a large number of studies have revealed a potential therapeutic strategy via the use of biomimetic nanoparticles. For instance, Zhu et al. created a delivery system consisting of BBR@PLGA@PLT nanoparticles by encapsulating berberine (BBR) within PLGA nanoparticles that are coated with platelet (PLT) membranes. They showed that BBR@PLGA@PLT nanoparticles facilitate the M2 polarization of macrophages in the infarcted myocardium, leading to beneficial outcomes, such as enhanced cardiac function, decreased collagen deposition in the heart, improved stiffness of scar tissue, and remarkable effects on angiogenesis [137]. Torrieri et al. introduced a synergistic nanosystem composed of pH-sensitive Putre-AcDEX nanoparticles (NPs) that incorporate atrial natriuretic peptide (ANP) and the p32-targeting linear TT1 peptide, which encapsulate two compounds that promote cellular self-renewal, namely CHIR99021 and SB203580. The Putre-AcDEX-PEG-TT1-ANP NPs demonstrated the ability to lower CD86 expression in M1-like macrophages, suggesting that this nanosystem possesses anti-inflammatory characteristics and may offer therapeutic benefits for MI [138]. Hu et al. created a mannan-based nanomedicine, Que@MOF/Man, designed for the targeted delivery of the antioxidative and anti-inflammatory compound quercetin (Que) to damaged myocardial tissue and activated macrophages in a controlled fashion. Consequently, Que@MOF/Man has been shown to significantly promote the differentiation of macrophages towards a reparative M2 phenotype. Considering the notable recruitment of macrophages in the myocardium following MI, Que@MOF/Man presents itself as a promising option for a therapeutically targeted approach in the context of MI [139]. Li et al. developed a biomimetic nanoparticle for Tregs (CsA@PPTK) by covering the nanoparticle with the membrane of platelets. This nanoparticle demonstrated a notable ability to scavenge ROS and enhanced both the production of Tregs and the ratio of M2 macrophages to M1 macrophages within the ischemic heart tissue. Additionally, CsA@PPTK significantly decreased cardiomyocyte apoptosis and minimized both the size of the infarct and the fibrotic area in the ischemic myocardium [140]. Mo et al. developed PEGylated allomelanin nanoparticles (AMNPs@PEG) to enhance the therapeutic effectiveness in cases of myocardial ischemia/reperfusion (I/R) injury. The AMNPs@PEG facilitated the transition of macrophages from the M1 to M2 subtype and diminished neutrophil recruitment. This shift was associated with a lower expression of pro-inflammatory genes while simultaneously increasing the levels of anti-inflammatory genes. Consequently, this led to a significant decrease in the size of the myocardial infarction and a notable enhancement in cardiac function following MI [141]. Li et al. developed a biomimetic nanosystem (Pd@CeO_2_-M), which is composed of exterior macrophage-derived extracellular vesicles (MEVs) and encapsulated Pd@CeO_2_ heterostructures. The Pd@CeO_2_ heterostructures exhibited outstanding inflammation-targeting ability and suppressed inflammatory response as well as facilitates macrophage polarization to M2 phenotype, and ultimately improved cardiac function and ventricular remodeling [142]. Zhu et al. developed a biomimetic nano-buffer system consisting of VEGF-encapsulated nanoparticles coated with *N*-acetyl-l-cysteine (NAC)-modified macrophage membrane, and found that V/P@M-NAC nanoparticles facilitated the polarization of macrophages toward the M2 phenotype, which promoted the maturation of new blood vessels in the infarcted myocardium [143]. Wang et al. engineered an anti-pyroptosis biomimetic nanoplatform (NM@PDA@PU), employing polydopamine (PDA) nanoparticles enveloped with a neutrophil membrane (NM) for the targeted delivery of puerarin (PU). The targeted delivery of puerarin could reshape the inflammatory microenvironment by reprogramming macrophages into the anti-inflammatory M2 subtype [144]. Lei et al. developed a platelet membrane (PM)-encapsulated baicalin nanocrystalline (BA NC) nanoplatform with a high drug load, BA NC@PM, which co-targets endothelial cells and macrophages through the transmembrane proteins of the PM to promote angiogenesis and achieve anti-inflammatory effects. An in vitro assay manifested that BA NC@PM could significantly reduce the levels of pro-inflammatory factors in the plasma and heart of MI/RI mice and promote macrophage polarization from the M1 to the M2 phenotype [145]. Xu et al. developed a platelet (PLT) membrane nanocarrier (PL720) that encapsulates l-arginine and FTY720 to facilitate the cascade-targeted delivery of these substances to the myocardial injury site and enable the controlled release of l-arginine and FTY720. During the late reperfusion inflammatory phase, PL720 promotes M2 macrophage polarization, thereby exhibiting anti-inflammatory and accelerated repair effects [146].

Furthermore, recent progress in developing cardiac patches has opened a new hope for the treatment of MI. Zhu et al. developed a patch that is functionalized with nitrates, where nitrate pharmacological groups are covalently attached to biodegradable polymers, thereby converting small molecule medications into therapeutic biomaterials. The delivery of nitric oxide through these nitrate-functionalized cardiac patches influence the polarization of macrophages towards the M2 phenotype, which is known to enhance wound healing and the process of tissue repair [147]. Similarly, Li et al. showed that an engineered cardiac patch known as AAB has the capability to boost regenerative M2 macrophage activity and reduce the inflammatory polarization of macrophages by decreasing intracellular ROS levels [148]. Coincidentally, in 2024, Xiao investigated the effect of a 2-Deoxy-Glucose (2-DG)-loaded chitosan/gelatin composite patch which significantly inhibited the expression of inflammatory cytokines, alleviated reactive oxygen species (ROS) accumulation, repressed the pro-inflammatory polarization of macrophages, attenuated local inflammatory microenvironment in the ischemic hearts, improved cardiac function, reduced scar size, and promoted angiogenesis post-MI [149]. Researchers Lee and colleagues formulated a hydrogel patch that is both paintable and adhesive, utilizing dextran–aldehyde (dex-ald) and gelatin. This hydrogel was designed to include the anti-inflammatory protein ANGPTL4, which releases gradually into the infarcted heart to help reduce inflammation. The application of ANGPTL4-infused hydrogel patches on heart tissues demonstrated enhanced vascular growth, decreased inflammatory macrophage presence, resulted in better repair of heart tissue, and caused the maturation of cardiac cell structures [150]. In addition, studies showed that major histocompatibility complex class II (MHC-II)^low^ macrophages play essential roles in tissue repair. Hao et al. developed and constructed a bi-laminated cardiac patch utilizing ECM materials infused with extracellular vesicles (EVs) sourced from mesenchymal stromal cells. Subsequent investigations revealed that applying EV-ECM patches to the infarcted region elevated the presence of immunomodulatory MHC-II^lo^ macrophages during the initial phase of myocardial damage, thereby reducing the excessive inflammatory reactions resulting from the injury [151]. Sigaroodi et al. presented an innovative bilayer nanofiber cardiac patch made from polycaprolactone (PCL), poly (xylitol sebacate) (PXS), and multiwalled carbon nanotubes (MWCNTs). This cardiac patch facilitates a conducive microenvironment for myocardial healing by diminishing the inflammatory responses in ischemic tissue and enhancing the M2/M1 macrophage ratio [152]. Nakkala et al. found that the poly-ε-caprolactone/itaconate-derivative dimethyl itaconate (PCL/DMI) nanofiber patches show excellent myocardial protection by modulating the polarization of inflammatory M1 macrophages into alternatively activated M2 macrophages in vitro, and by enhancing the functions of the left ventricle and reducing the activity of genes associated with inflammation in vivo [153]. He et al. innovatively constructed a two-layer asymmetric Janus hydrogel patch, with the top layer having anti-cell adhesion properties while the bottom hydrogel layer can stably adhere to cardiac tissue and be removed on demand. The results indicate that in the experimental group treated with Janus hydrogel, CD68-positive cells decreased while CD206 cells increased, suggesting that macrophages mainly exist in the M2 phenotype and demonstrating excellent anti-inflammatory effects. This multifunctional Janus hydrogel patch holds great promise in overcoming the limitations of current cardiac patches [154]. Lee et al. synthesized a hydrogel cardiac patch designed to offer mechanical support, facilitate electrical conduction, and enhance tissue adhesion, all of which contribute to the restoration of function in an infarcted heart. Both in vivo and in vitro studies demonstrated that the CAH cardiac patch prompted macrophages to shift toward an M2 phenotype, which helped reduce the inflammatory response in the affected region and prevented cell apoptosis and necrosis [155]. Wei et al. designed a new type of cardiac patch, which was composed of stem cell layers derived from brown adipose tissue and conductive electrospun nanofibers (CPSNs) doped with multiwalled carbon nanotubes. The hybrid cardiac patches demonstrated enhanced angiogenesis and reduced inflammation in a rat model of MI. Furthermore, the CNBS cardiac patches had the ability to direct macrophages towards M2 polarization and facilitate the remodeling of gap junctions, thereby aiding in the restoration of cardiac functions [156]. All those aforementioned engineered cardiac patches have the potential to eventually alleviate inflammation and enhance angiogenesis following MI, subsequently restoring heart function and offering a promising therapeutic approach for the treatment of MI.

Last but not least, as highly efficient natural drug carriers, extracellular vesicles have been demonstrated to significantly improve the therapeutic effects on myocardial infarction. Xiao et al. developed an extracellular vesicle-based delivery platform for transferring growth differentiation factor-15 (GDF15) and discovered that EXO-GDF15 played a crucial role in modulating the phenotypic transformation of macrophages, reducing apoptosis in cardiomyocytes, and promoting angiogenesis in endothelial cells. Additionally, EXO-GDF15 also notably influenced macrophage diversity and inflammatory cytokine levels, diminished the fibrotic region, and improved cardiac performance in rats post-infarction [157]. Chen et al. developed cell membrane-modified extracellular vesicles (MmEVs) loaded with thymosin β4 (Tβ4) which could skew macrophages towards the pro-healing M2 type, thus promoting cardiomyocyte proliferation and endothelial cell migration [158]. Ning et al. reported that EVs derived from bone marrow mesenchymal stem cells (MSCs) pretreated with atorvastatin (ATV) (MSCATV-EV) transfer miR-139-3p significantly reduced the amount of CD68^+^ total macrophages and increased the amount of CD206^+^ M2 macrophages, remarkably alleviating inflammation and enhancing the efficacy of cardiac repair for AMI [159]. Gong et al. found that exosomes derived from nicorandil-pretreated MSC (MSCNIC-exo) promote macrophage M2 polarization by upregulating the miR-125a-5p-targeting TRAF6/IRF5 signaling pathway, which has great potential for facilitating cardiac repair post-infarction [160].

The aforementioned studies targeting monocytes/macrophages indicated the two predominant strategies using different delivery systems to control the balance mediated by macrophages, specifically by preventing the excessive accumulation of pro-inflammatory macrophages and by altering the phenotypes of macrophages.

### 4.3. Targeting Lymphocytes

Regulatory T cells are key immune regulators in the ischemic myocardium and possess anti-inflammatory properties [161]. A report indicated that rapidly increasing Treg number in the circulation post-MI via the systemic administration of exogenous Tregs improves cardiac function [162]. Concurrently, suppressing the recruitment of Tregs aggravates myocardial injury [163]. Thus, enhancing the presence of Tregs within ischemic myocardial tissue represents a potent therapeutic approach. Zhang et al. created a drug delivery system that utilizes alginate hydrogel for the continuous release of cell-derived exosomes (DEXs). They discovered that DEXs-Gel markedly improved the therapeutic outcomes of DEXs in terms of enhancing cardiac function following MI. Flow cytometry along with immunofluorescence staining demonstrated that DEXs notably increased the infiltration of Treg cells and M2 macrophages into the peri-infarct zone after MI [164].

Gao et al. developed an injectable drug-releasing microgel system (MTK-TK-drug) capable of responding to stimuli from reactive oxygen species (ROS) using a coaxial capillary microfluidic approach combined with UV curing techniques. These MTK-TK-drug microgels were effective in transforming pro-inflammatory Th17 cells into anti-inflammatory regulatory T cells (Treg) in vitro. Additionally, the materials that scavenge ROS worked together to enhance this effect by influencing the inflammatory microenvironment. As a result, the microgels significantly mitigated cardiomyocyte apoptosis and reduced the inflammatory response following myocardial infarction (MI) in vivo, which contributed to decreased fibrosis, improved vascularization, and aided in the preservation of cardiac function [165]. Wang et al. prepared a Treg-regulated microgel scaffold (GTK-TK-drug) loading with pro-drug aminooxyacetic acid (AOA). Immunofluorescence staining revealed the effectiveness of AOA for the differentiation of T cells into Tregs, and the upregulation of Tregs significantly enhanced the secretion of anti-inflammatory TGF-β1 and inhibited the secretion of pro-inflammatory TNF-α and IL-6 factors [166]. In addition, liposomal nanoparticles containing MI antigens and rapamycin (l-Ag/R) facilitate the efficient generation of antigen-specific regulatory T cells (Tregs). Notably, administering l-Ag/R intradermally in mice with acute MI reduces myocardial inflammation by promoting Tregs and shifting macrophage polarization from inflammatory to reparative, thereby mitigating negative cardiac remodeling and enhancing cardiac function [167].

Additionally, decreasing the recruitment of T cells emerges as a robust therapeutic approach. Tortajada et al. developed a cardiac decellularized scaffold infused with peptide hydrogel to deliver EVs derived from porcine cardiac adipose tissue-derived mesenchymal stem cells (cATMSC). In vivo experiments indicated that the nanovesicle cATMSC-EVs facilitated a rise in vascular density and a decrease in T cell infiltration in the injured myocardium [168].

### 4.4. Targeting Fibroblast

There is a report which indicated that exosomes from human UMSCs promote the differentiation of fibroblasts toward myofibroblasts, thus attenuating the inflammatory response, reducing cardiomyocyte apoptosis, and promoting cardiac repair [169]. Hence, a key approach to targeting fibroblasts involves modulating their cellular phenotype. Wang et al. developed biomimetic nanoparticles that are distinguished by a combination of cell membranes derived from both MSCs and macrophages incorporating miR-125b. The miR-125b-MRCNPs exhibited anti-inflammatory properties conferred by the MSCs membrane proteins along with miR-125b. By synergistically combining ultrasound-targeted microbubble destruction (UTMD) technology with the inherent inflammatory chemotaxis capabilities of cell membranes, these nanoparticles effectively delivered their payload to the inflamed regions. This process regulates various target factors, preventing cardiomyocyte apoptosis, restricting fibroblast proliferation, and ultimately facilitating the recovery of myocardial function after MI, while ensuring precise and targeted therapeutic effects [170]. Wen et al. created nanoparticles designed to home in on the damaged heart and gradually deliver an MCP-1-binding peptide, HSWRHFHTLGGG (HSW), which counteracts the elevated MCP-1 levels. Their research demonstrated that HSW decreased the migration of monocytes, inhibited the increase in pro-inflammatory cytokines, and curtailed the differentiation of myofibroblasts in vitro [171]. Wang et al. observed that exosomes derived from cardiomyocytes are crucial in the phenoconversion of cardiac myofibroblasts after MI. The miR-92a is delivered to fibroblasts as part of the exosomal content, alleviating the SMAD7-mediated suppression of αSMA transcription, which in turn activates cardiac myofibroblasts [172].

Therefore, we summarize the recent progress using the state-of-the-art delivery systems to target various cells involved in inflammation in MI (Table 1), especially neutrophiles and monocytes/macrophages, by controlling the infiltration and modulating the phenotypes of these cells post-MI, which provide effective approaches to balancing inflammation and promoting cardiac repair.

## 5. Conclusions

In this review, we introduced the recruitment and differentiation of major inflammatory cells in MI. Furthermore, the latest delivery systems have been explored. In addition, this review mainly summarizes the principal inflammation modulation strategies and current applications of the emerging delivery systems targeting inflammatory cells in the treatment of MI. By enhancing the comprehension of the diverse functions of immune cells and fibroblasts, alongside acquiring significant insights into the simultaneous advancement of drug delivery systems, this research is likely to inspire more targeted and precise anti-inflammatory therapies, promote efficient cardiac repair, and ultimately elevate patients’ quality of life while alleviating the societal healthcare burden. However, current knowledge on myocardial infarction is largely based on studies in mouse and rat models, with limited studies examining immune cell or fibroblast recruitment post-MI in humans, which is expected to be explored in the future.

## Figures and Tables

**Figure 1 bioengineering-12-00205-f001:**
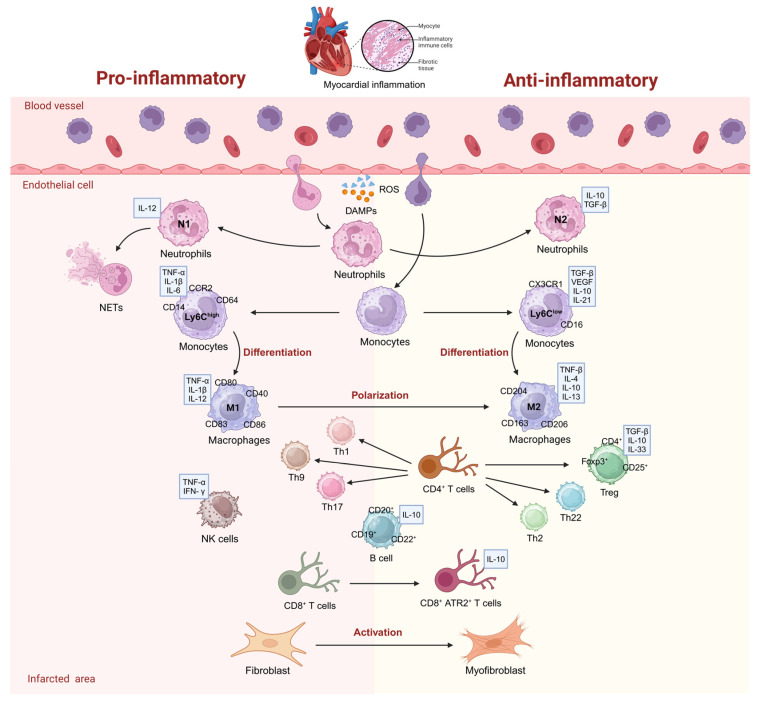
The inflammatory response cells involved after myocardial infarction.

**Figure 2 bioengineering-12-00205-f002:**
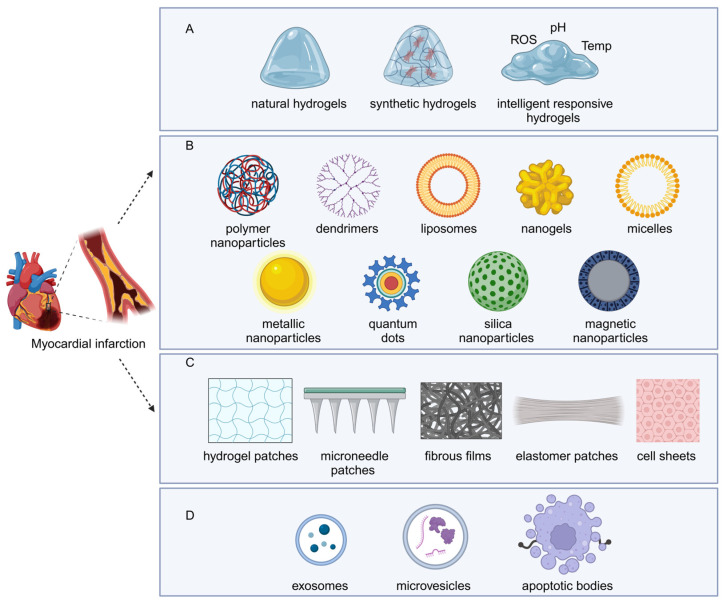
Overview of the delivery system profile. (**A**) The classification of hydrogels; (**B**) The classification of nanoparticles; (**C**) The classification of cardiac patches; (**D**) The classification of extracellular vesicles.

**Table 1 bioengineering-12-00205-t001:** Inflammatory cell-targeting strategies for MI treatment.

Targets	Delivery Systems	Therapeutic Agents	References
Neutrophils	Alleviate the infiltration of neutrophils		
	Hydrogel	Exosome	[93]
	Nanoparticle	Azithromycin	[94]
	Nanoparticle	/	[95]
	Nanoparticle	siR-ICAM1 and pCXCL12	[96]
	Nanoparticle	IL-5	[97]
	Nanoparticle	siVCAM-1 and DXM	[98]
	Cardiac patch	miR-29b	[99]
	Extracellular vesicle	miR-199	[100]
	Apoptosis of activated neutrophils		
	Nanoparticle	Roscovitine	[101]
	Promote N2 neutrophil polarization		
	Injectable hydrogel	Cell	[102]
Monocytes	Decrease the recruitment of monocytes		
	Injectable hydrogel	Irbesartan	[103]
	Nanoparticle	Cyclosporine A or pitavasatin	[104]
	Extracellular vesicle	miR-24-3p	[105]
	Interact with monocytes		
	Nanoparticle	/	[106]
	Nanoparticle	Resolvin D1	[107]
	Nanoparticle	Notoginsenoside R1	[108]
	Nanoparticle	miR-21	[109]
Macrophages	Regulate the infiltration of macrophages		
	Injectable Hydrogel (thermoresponsive)	VEGF, IL-10 and PDGF	[110]
	Injectable hydrogel	SPIO	[111]
	Injectable hydrogel (conductive)	OGGP3	[112]
	Injectable hydrogel	BASCs	[113]
	Nanoparticle	O_2_	[114]
	Cardiac patch	CA and ITA	[115]
	Extracellular vesicle	miR-146a-5p	[116]
	Extracellular vesicle	CMP	[117]
	Inhibit M1 macrophages polarization		
	Injectable hydrogel	ADSC-Exos	[118]
	Injectable hydrogel	miR-21-5p	[119]
	Injectable hydrogel	Puerarin and chitosan	[120]
	Nanoparticle	Atorvastatin calcium	[121]
	Extracellular vesicle	miR-25-3p	[122]
	Promote pro-inflammatory M1 macrophages polarize into pro-healing M2 macrophages		
	Injectable hydrogel	AACs and VEGF	[123]
	Injectable hydrogel (MMP responsive)	CTL4	[124]
	Injectable hydrogel	Stem cells	[125]
	Injectable hydrogel	Rosmarinic acid	[126]
	Injectable hydrogel	Curcumin and collagen type Ⅲ	[127]
	Hydrogel	Triptolide	[128]
	Hydrogel	/	[129]
	Injectable hydrogel	Fisetin	[130]
	Injectable hydrogel	/	[131]
	Injectable hydrogel	Quercetin	[132]
	Injectable hydrogel	Exosome	[133]
	Injectable hydrogel	Oncostatin	[134]
	Injectable hydrogel	IL-10 and SN50	[135]
	Injectable hydrogel	Annexin A1	[136]
	Nanoparticle	Berberin	[137]
	Nanoparticle	CHIR99021 and SB203580	[138]
	Nanoparticle	Quercetin	[139]
	Nanoparticle	/	[140]
	Nanoparticle	/	[141]
	Nanoparticle	/	[142]
	Nanoparticle	VEGF	[143]
	Nanoparticle	Puerarin	[144]
	Nanoparticle	/	[145]
	Nanoparticle	PL720	[146]
	Cardiac patch	Nitric oxide	[147]
	Cardiac patch	/	[148]
	Cardiac patch	2-Deoxy-Glucose	[149]
	Cardiac patch	ANGPTL4	[150]
	Cardiac patch	MSC EVs	[151]
	Cardiac patch	MWCNT	[152]
	Cardiac patch	Dimethyl itaconate	[153]
	Cardiac patch	/	[154]
	Cardiac patch	/	[155]
	Cardiac patch	/	[156]
	Extracellular vesicle	GDF-15	[157]
	Extracellular vesicle	Thymosin β4	[158]
	Extracellular vesicle	miR-139-3p	[159]
	Extracellular vesicle	miR-125a-5p	[160]
Lymphocytes	Regulate the phenotype of lymphocytes		
	Hydrogel	DEXs	[164]
	Injectable hydrogel	/	[165]
	Hydrogel	Aminooxyacetic acid	[166]
	Nanoparticle	/	[167]
	Extracellular vesicle	/	[168]
Fibroblasts	Promote the activation of fibroblasts into myofibroblasts		
	Nanoparticle	miR-125b	[170]
	Nanoparticle	HSWRHFHTLGGG	[171]
	Extracellular vesicle	miR-92a	[172]

## Data Availability

Data sharing is not applicable.

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
