# Peer review of "Inflammatory Cell-Targeted Delivery Systems for Myocardial Infarction Treatment"

_bioengineering, 2025, doi:10.3390/bioengineering12020205_

Round 1

Reviewer 1 Report

Comments and Suggestions for Authors

It is recommended for publication. It is good as it is and I do not have any additional comments to make.

Author Response

Comments 1: It is recommended for publication. It is good as it is and I do not have any additional comments to make.

Response 1: Thank you very much for your positive feedback and recommendation!

Reviewer 2 Report

Comments and Suggestions for Authors

Your MS is significant for MI research. Please correct some text style errors. Please use Italics font for et al., as well for in vitro. In line 266 use drug instead Drug. Reference N6 article name express as in reference N1; 2 a.o. Please  omit colored (green, red) marks in the final version of the MS text.

Author Response

Comments 1: Please correct some text style errors.

Response 1: Thank you for pointing this out, I have reviewed full article and revised some text style, including line 57, changed “damaged” into “damage”; line 360, changed “Vesicles” into “vesicles”; line 164, changed “For insatnce” into “For instance”; line 457, remove the excessive periods at the end of the sentence; line 860, changed “Targeting Fibroblast” into “Targeting fibroblast”;

Comments 2: Please use Italics font for et al., as well for in vitro.

Response 2: Yes, I have revised all of et al., as well for in vitro into Italics.

Comments 3: Reference N6 article name express as in reference N1

Response 3: Dear Reviewer, we have carefully checked and ensured that all references are formatted in ACS style.

Comments 4: In line 266 use drug instead Drug.

Response 4: Yes, I have changed it.

Comments 5: Please omit colored (green, red) marks in the final version of the MS text.

Response 5: Certainly! I have omitted all the colored marks in the final version of the manuscript. Thank you for your guidance.

Reviewer 3 Report

Comments and Suggestions for Authors

The manuscript by  Liu et al on the use of Cell Targeted Delivery Systems for Myocardial Infarction Treatment is a well written review in an area of strong bio-medical interest.

The abstract and text are well written and easy to understand.

The two figures and the table add to the review and aid understanding.

Referencing is good.

Publish as is.

Author Response

Comments 1: The manuscript by Liu et al on the use of Cell Targeted Delivery Systems for Myocardial Infarction Treatment is a well written review in an area of strong bio-medical interest.

The abstract and text are well written and easy to understand.

The two figures and the table add to the review and aid understanding.

Referencing is good.

Response 1: Thank you very much for your positive evaluation and comments.